# The neuropeptide sulfakinin is a peripheral regulator of insect behavioral switch between mating and foraging

Hong-Fei Li[1,2], Bao Dong[1,2], Yuan-Yuan Peng[1,2], Hao-Yue Luo[1,2], Xiao-Lan Ou[1,2], Zheng-Lin Ren[1,2], Yoonseong Park[3]*, Jin-Jun Wang[1,2]*, Hong-Bo Jiang[1,2]*

[1]Key Laboratory of Entomology and Pest Control Engineering, College of Plant Protection, Southwest University, Chongqing, China; [2]Key Laboratory of Agricultural Biosafety and Green Production of Upper Yangtze River (Ministry of Education), Academy of Agricultural Sciences, Southwest University, Chongqing, China; [3]Department of Entomology, Kansas State University, Manhattan KS, United States

*For correspondence:
ypark@ksu.edu (YP);
wangjinjun@swu.edu.cn (J-JW);
jhb8342@swu.edu.cn (H-BJ)

Competing interest: The authors declare that no competing interests exist.

## eLife Assessment

This **important** work investigates the mechanism that underlies the switch between feeding and mating behaviors in the oriental fruit fly, *Bactrocera dorsalis*. Using a variety of approaches, the authors show that this switch is mediated by the neuropeptide, sulfakinin, acting peripherally through the sulfakinin receptor 1 to regulate the expression of antennal odorant receptors. The evidence is **solid** in support of the hypothesis that sulfakinin signaling mediates changes in the periphery, although additional sites of action may also contribute to these changes.

**Abstract** Behavioral strategies for foraging and reproduction in the oriental fruit fly (*Bactrocera dorsalis*) are alternative options for resource allocation and are controlled by neuropeptides. Here, we show that the behavioral switch between foraging and reproduction is associated with changes in antennal sensitivity. Starved flies became more sensitive to food odors while suppressing their response to opposite-sex pheromones. The gene encoding sulfakinin receptor 1 (*SkR1*) was significantly upregulated in the antennae of starved flies, so we tested the behavioral phenotypes of null mutants for the genes encoding the receptor (*skr1*[–/–]) and its ligand sulfakinin (*sk*[–/–]). In both knockout lines, the antennal responses shifted to mating mode even when flies were starved. This suggests that sulfakinin signaling via SkR1 promotes foraging while suppressing mating. Further analysis of the mutant flies revealed that sets of odorant receptor (OR) genes were differentially expressed. Functional characterization of the differentially expressed ORs suggested that sulfakinin directly suppresses the expression of ORs that respond to opposite-sex hormones while enhancing the expression of ORs that detect food volatiles. We conclude that sulfakinin signaling via SkR1, modulating OR expressions and leading to altered antenna sensitivities, is an important component in starvation-dependent behavioral change.

## Introduction

Foraging and mating behaviors are alternative options for resource allocation in insects. Starved and reproductively immature individuals intensify their foraging activity, whereas satiated and reproductively mature individuals increase their mating activity to facilitate reproduction (*Chen et al., 2023*; *Cheriyamkunnel et al., 2021*; *Ebrahim et al., 2023*; *Ren et al., 2021*). Such behavioral changes involve complex neural pathways triggered by neuropeptide signaling (*González-Segarra et al.,*

*2023*; *Nässel and Zandawala, 2022*; *Schoofs et al., 2017*; *Sun et al., 2023*; *Zhang et al., 2022b*; *Zhang et al., 2022a*). The key regulators have been identified in the fruit fly *Drosophila melanogaster*, including the neuropeptides sulfakinin (Sk) (*Wang et al., 2022*; *Wu et al., 2019*) and short neuropeptide F (sNPF) (*Ko et al., 2015*; *Root et al., 2011*). Tyramine and diuretic hormone 31 then resolve the conflict between foraging and mating behavior in the central nervous system (*Cheriyamkunnel et al., 2021*; *Zhang et al., 2022b*).

In mammals, cholecystokinin (CCK) transmits satiety by activating cholecystokinin receptor 1, leading to the expression of genes encoding other anorectic and orexigenic peptides and receptors in vagal neurons (*Beutler et al., 2017*; *Burdyga et al., 2008*; *Essner et al., 2017*). CCK is also involved in the central integration of sensory inputs to regulate sexual behavior (*Bloch et al., 1987*; *Micevych et al., 1988*). In insects, Sk, the ortholog of CCK, and its signals via two receptors, SkR1 (CCKLR-17D3 of *D. melanogaster*) and SkR2 (CCKLR-17D1), regulate sub-behavioral steps during foraging, food consumption, and mating in *D. melanogaster* (*Fedina et al., 2023*; *Wang et al., 2022*). For example, Sk suppresses the expression of the *Gr64f* gustatory receptor gene, which inhibits feeding as a signal for the onset of satiety (*Guo et al., 2021*). Sk also inhibits both male and female sexual behavior (*Wu et al., 2019*). However, the complexity of Sk pathways, spatial and temporal dynamics of multiple ligands and receptors, is exemplified by cell-specific signaling via SkR1 for the promotion of mating receptivity in virgin females (*Wang et al., 2022*), which is an opposite action of the inhibition of sexual behavior (*Wu et al., 2019*). Modulation of peripheral sensory inputs, including odorant perception, is involved in the control of mating and foraging behavior in addition to the central neuronal circuits.

The oriental fruit fly, *Bactrocera dorsalis,* is a destructive agricultural pest insect. Control of *B. dorsalis* includes the use of the reproductive and feeding behavioral responses to semiochemicals, such as the components in the sex pheromones and in the food smell (*Jaffar et al., 2023*). Chemical components in the sex pheromone of this species includes 2,3,5-trimethyl pyrazine, 2,3,5,6-tetramethylpyrazine, ethyl laurate, ethyl myristate, ethyl *cis*-9-hexadecenoate, and ethyl palmitate (*Chen et al., 2023*; *Ren et al., 2021*) and in the food smells are ethyl benzoate, ethyl butyrate, and methyl eugenol, among others. (*Kamala Jayanthi et al., 2014*; *Zhu et al., 2018*). The physiology and molecular mechanisms underlying the behavioral modulation involved in the different types of chemical attractants have not yet been studied in this species of insect.

We hypothesized that neuropeptide Sk-SkR and other neuropeptidergic systems are involved in the switch between foraging success and mating behaviors in the *B. dorsalis*. We were specifically interested in the alteration in odorant perceptions in the antennae leading to the success in foraging and/or mating depending on the physiological status. Sequential approaches, starting with the unbiased test for involvement of odorant perception in the behavioral switch, led us to further in-depth investigations of the role of neuropeptide and Sk-SkR systems. We demonstrate that sulfakinin signaling via SkR1 in the antennae alters the expression of sets of odorant receptor (OR) genes during starvation enhancing the foraging success rate. Our results thus provide insight into sulfakinin signal on peripheral olfactory system that is coordinated with the central neuronal circuits for modulation of behavior.

## Results

### Altered antennal sensitivity is associated with starvation-induced foraging and satiation-induced mating success

We developed an assay to investigate the behavioral switch between foraging and mating induced by the physiological state of *B. dorsalis* (*Figure 1A and B*). Foraging success increased with the duration of the starvation period (*Figure 1D*, *Figure 1—figure supplement 1*), whereas courtship and copulation rates decreased, also with the starvation length-dependent manner (*Figure 1G and H*). Starved flies tended to move directionally toward the food source located in the center of the arena (*Figure 1C*).

We therefore tested antennal sensitivities toward artificial food and opposite-sex body extracts using an electroantennogram (EAG). Flies of both sexes starved for 12 h showed a greater EAG response to food odor (*Figure 1E and F*) but a weaker response to opposite-sex body extracts (*Figure 1I and J*). Furthermore, starved flies also showed greater EAG responses to food volatile components and weaker responses to sex pheromone components (*Figure 1—figure supplement*

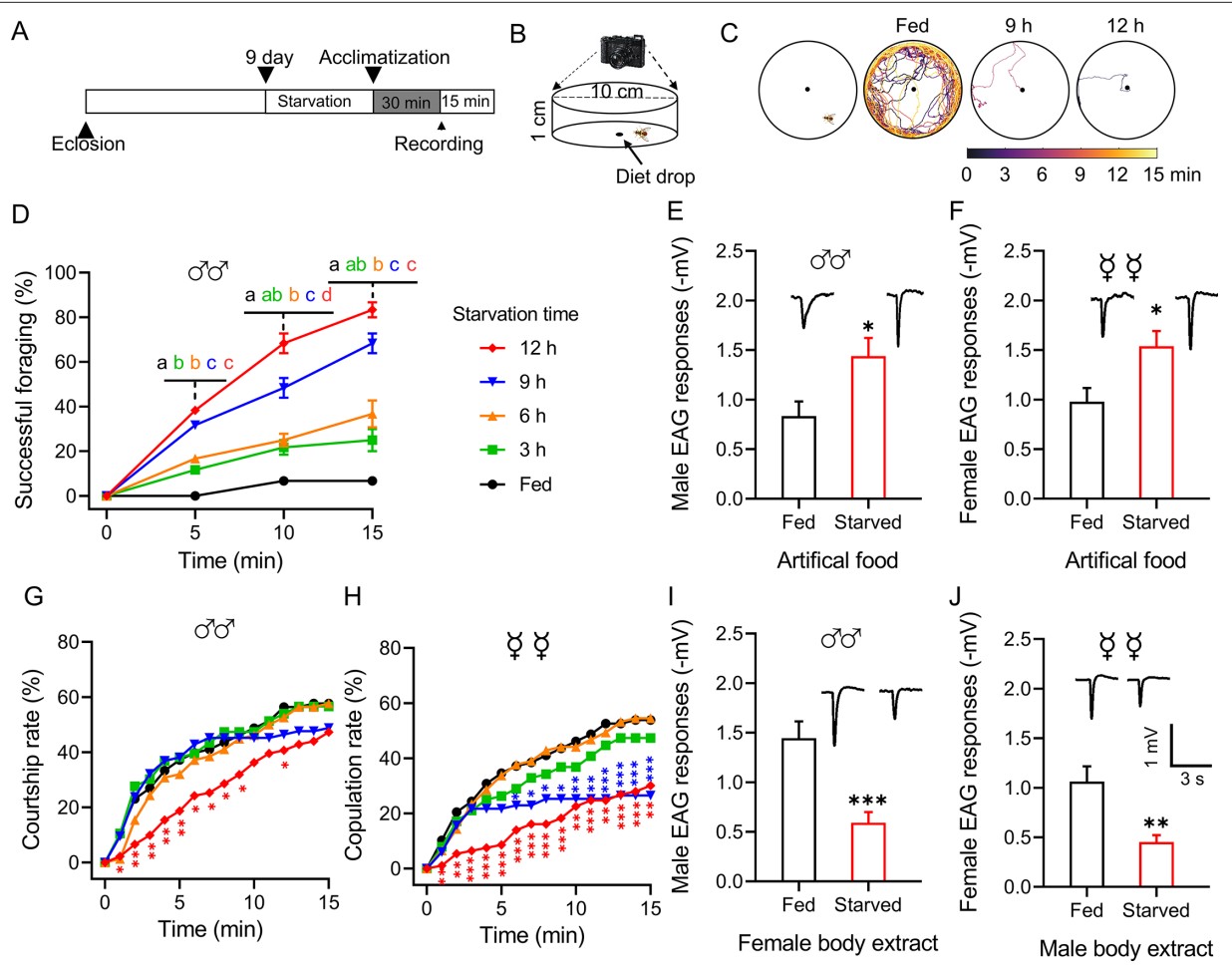

**Figure 1.** Starvation alters the behavior and olfactory responses of *B. dorsalis*. (**A**) Schematic illustration of the experimental design. (**B**) Schematic illustration of the mating behavior assay device. (**C**) Representative foraging trajectories in the 100 mm diameter arenas within a 15 min observation period of flies starved for different durations. (**D**) Cumulative successful foraging within a 15 min observation period of male flies starved for different durations. Data are means ± SEM, n=60 flies for each condition. Different letters above the error bars indicate significant differences (one-way ANOVA followed by Tukey's multiple comparisons test; p<0.05). (**E, F**) Electroantennogram (EAG) responses to artificial food in fed and starved male (**E**) and female (**F**) flies (unpaired *t*-test). (Top) Representative EAG recordings. (**G**) Cumulative courtship rate within a 15 min observation period of flies starved for different durations (n=78, 76, 78, 84, and 91, respectively, from fed to 12 h). (**H**) Cumulative copulation rate within a 15 min observation period of the flies starved for different durations (n=78, 76, 79, 83, and 93, respectively, from fed to 12 h). (**G, H**) Kruskal–Wallis and *post hoc* Mann–Whitney *U* tests were applied. (**I, J**) EAG responses to body extracts in fed and starved male (**I**) and female (**J**) flies. (Top) Representative EAG recordings. Data are means ± SEM, n=10–12 antennae per genotype (unpaired *t*-test; *p<0.05, **p<0.01, ***p<0.001).

The online version of this article includes the following source data and figure supplement(s) for figure 1:

**Source data 1.** Starvation alters the behavior and olfactory responses of *B. dorsalis*.

**Figure supplement 1.** The cumulative successful foraging within a 15 min observation period of female flies with different starvation durations.

**Figure supplement 1—source data 1.** The cumulative successful foraging within a 15 min observation period of female flies with different starvation durations.

**Figure supplement 2.** Electroantennogram (EAG) responses of fed and starved flies to food relevant odors and sex pheromones at three concentrations.

**Figure supplement 2—source data 1.** EAG responses of fed and starved flies to food relevant odors and sex pheromones at three concentrations.

*2*). This suggested antenna-mediated olfaction is associated with and required for successful foraging and mating.

## Starvation-induced *SkR1* mRNA in the antennae is required for the switch from mating to foraging behavior

To identify signaling molecules responsible for the olfactory-dependent behavioral switch, we compared the expression profiles of neuropeptide signaling components in the antennae of fed and starved flies. We found that the components of three neuropeptide signaling systems were significantly upregulated in the antennae of starved flies, namely *short neuropeptide F receptor* (*sNPFR*), *SIFamideR1*, and *SkR1*, the latter showing the strongest induction (*Figure 2A*). *Sk* did not show the expression change in the antennae, but significantly increased in the head in comparison between those in starvation and after re-feeding, suggesting that the circulatory Sk released from the central nervous system could be the source for activation of SkR1 in the antenna (*Figure 2—figure supplement 1*).

We introduced a 7 bp deletion into the sulfakinin gene (*Sk*) using the CRISPR/Cas9 system, causing a frameshift that abolished the sequence region encoding two predicted mature peptides: Sk-1 and Sk-2 (*Figure 2B*, *Figure 2—figure supplement 2A and B*). We confirmed the absence of an anti-sulfakinin immunoreactive signal in the central nervous system of the *sk*⁻/⁻ flies (*Figure 2—figure supplement 2C*). Only *SkR1* was upregulated in the starved flies among two predicted Sk receptor genes *SkR1* and *SkR2* (*Figure 2A*). We therefore generated a *SkR1* null mutant (*skr1*⁻/⁻) by introducing a 187 bp deletion using CRISPR/Cas9, resulting in a frameshift that resulted in a premature stop codon (*Figure 2B*, *Figure 2—figure supplement 3A–C*). We confirmed the absence of the anti-SkR1 immunoreactive signal in the antennae of the *skr1*⁻/⁻ flies (*Figure 2—figure supplement 3D*).

Starved *sk*⁻/⁻ and *skr1*⁻/⁻ flies showed the same locomotor activity and body size as wild-type (WT) controls when we measured their velocity and body length (*Figure 2C*, *Figure 2—figure supplement 4*). However, both null mutants consumed more food than WT flies (*Figure 2D*), consistent with previous results in *D. melanogaster* (*Guo et al., 2021*). Interestingly, *sk*⁻/⁻ and *skr1*⁻/⁻ male flies consumed more food at all times compared to the WT. However, the corresponding mutant females consumed more food only at night. This may be due to the high basal feeding rate of females during the daytime, masking the increase in feeding in the knockout of Sk signaling. As shown in *Figure 2D*, WT females ate more food than males, and WT females consumed more food during the daytime. Another major difference between the null mutants and WT controls was that the mutant flies were less successful at foraging even after starvation, similar to the characteristics of satiated WT flies (*Figure 3A and B*). However, foraging was not completely abolished. In addition, starved mutant females (*Figure 3C*) and males (*Figure 3D*) showed high mating success rates, as same rates as fed WT flies. We tested the behavioral phenotypes of heterozygous mutant of *Sk* knockout flies. The results showed that foraging and mating behaviors of Sk heterozygous mutants were unaffected during starvation, suggesting the mutants are completely recessive (*Figure 3—figure supplement 1*).

We next tested antennal sensitivity to food odors and body extracts by EAG in each genotype using both fed and starved flies in each case. The starved *sk*⁻/⁻ and *skr1*⁻/⁻ flies showed significantly weaker responses to food odors compared to the starved WT flies, although the response of *sk*⁻/⁻ flies was slightly stronger than that of fed WT flies (*Figure 3E and F*, *Figure 3—figure supplement 2*), which was similar to the behavioral response for a successful foraging rate (*Figure 3A and B*). The antennae of starved *sk*⁻/⁻ and *skr1*⁻/⁻ flies showed the same EAG response to opposite-sex body extracts as fed WT flies (*Figure 3G and H*, *Figure 3—figure supplement 2*), which was significantly higher than that of starved WT flies.

## Sets of ORs are differentially expressed when comparing satiated/starved WT and *sk*⁻/⁻ flies

To gain insight into the molecular basis of alteration of olfactory sensitivities by the Sk-SkR1 system, we extracted RNA from the head epidermal tissues (including antennae) of fed and starved flies for sequencing (RNA-Seq). We compared starved WT and starved *sk*⁻/⁻ flies with satiated WT flies, focusing on the differentially expressed OR genes (*Figure 4A*). We identified three OR genes (*OR7a.4*, *OR7a.8,* and *OR10a*) that were upregulated in the starved WT flies but unchanged in starved *sk*⁻/⁻ flies compared to satiated WT controls, and two (*OR49a* and *OR63a*) that were downregulated in starved

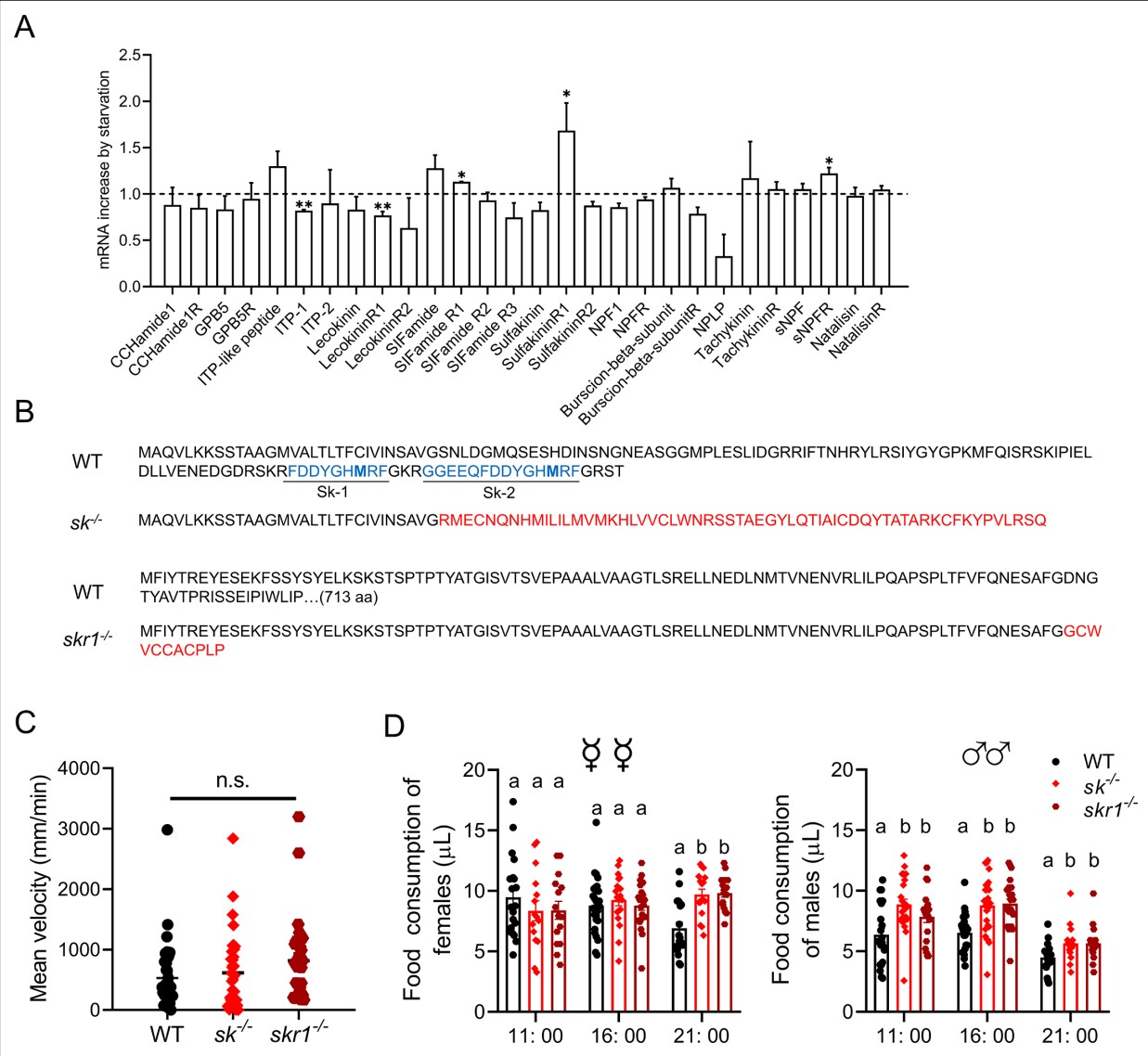

**Figure 2.** Starvation-induced expression of neuropeptide system components and phenotypes of null mutant flies. (**A**) Changes in the expression of neuropeptide system components in the antennae of starved flies. The relative expression values are the fold-changes compared to the control. Data are means ± SEM, n=3 (unpaired *t*-test; *p<0.05, **p<0.01). (**B**) Altered amino acid sequence and mature peptides in the null mutants *sk⁻/⁻* and *skr1⁻/⁻* compared to wild-type (WT) flies. (**C**) Mean velocity of flies representing each genotype during a 15 min observation period. Data are means ± SEM (Kruskal–Wallis test). (**D**) Knockout of sulfakinin and SkR1 increases food consumption in *B. dorsalis* at different times of day. Each experiment consisted of one fly, and at least 15 flies were contained for the assay. Different lowercase letters indicate significant differences between treatments (one-way ANOVA followed by Tukey's multiple comparisons test; p<0.05).

The online version of this article includes the following source data and figure supplement(s) for figure 2:

**Source data 1.** Starvation-induced expression of neuropeptide system components and phenotypes of null mutant flies.

**Figure supplement 1.** Refeeding after 12 h starvation decreases *SkR1* transcript.

**Figure supplement 1—source data 1.** Refeeding after 12 h starvation decreases *SkR1* transcript.

**Figure supplement 2.** Generation of the BdSk null mutant via CRISPR-Cas9 system.

**Figure supplement 3.** Generation of the *BdSkR1* null mutant via CRISPR-Cas9 system.

**Figure supplement 4.** Knockout of Sk did not affect the body length of *B. dorsalis*.

**Figure supplement 4—source data 1.** Knockout of Sk did not affect the body length of *B. dorsalis*.

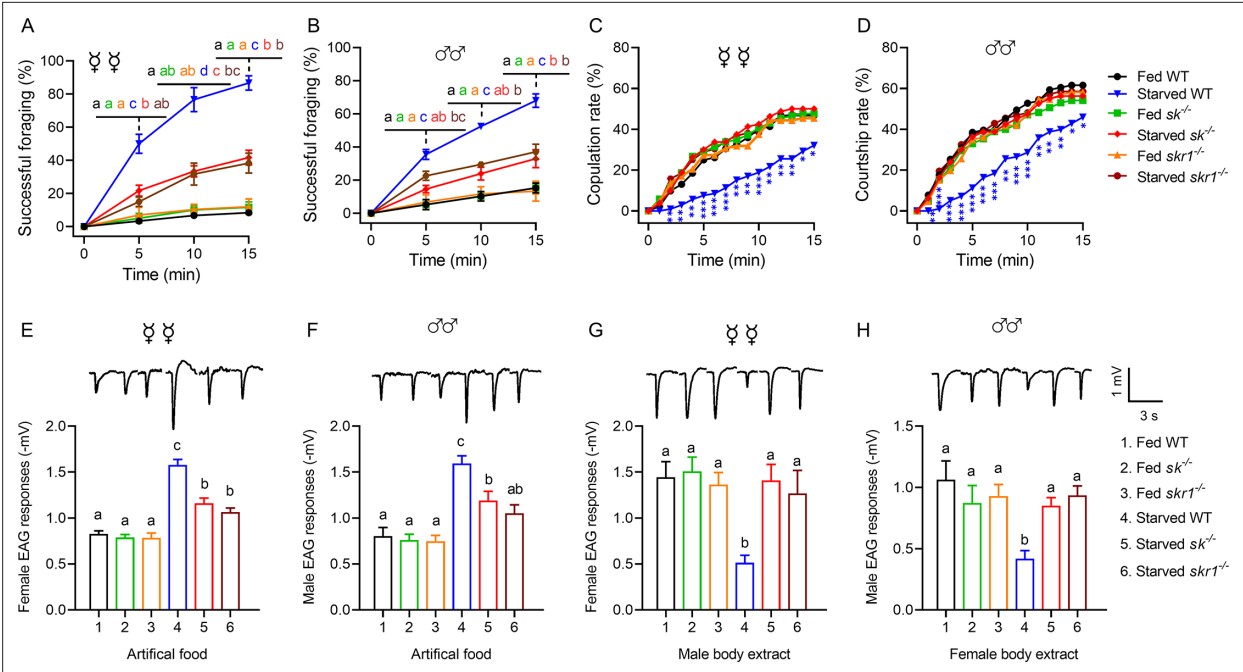

**Figure 3.** Sk-SkR1 signaling is required for changes in behavior and olfactory sensitivity induced by starvation. (**A, B**) Cumulative successful foraging within a 15 min observation period of fed and starved flies with different genotypes. The successful foraging of females (**A**) and males (**B**) was measured separately. Data are means ± SEM, n=60 flies for each condition. Different letters above the error bars indicate significant differences (one-way ANOVA followed by Tukey's multiple comparisons test; p<0.05). (**C**) Cumulative copulation rates within a 15 min observation period of fed and starved flies with different genotypes (n=92, 86, 88, 106, 80, and 76, respectively, from fed WT to starved *skr1⁻/⁻*). (**D**) Cumulative courtship rates within a 15 min observation period of fed and starved flies with different genotypes (n=91, 85, 86, 98, 80, and 77, respectively, from fed WT to starved *skr1⁻/⁻*). (**C, D**) Kruskal–Wallis and *post hoc* Mann–Whitney *U* tests were applied (*p<0.05, **p<0.01, ***p<0.001). (**E, F**) Electroantennogram (EAG) responses to artificial food in fed and starved flies of different genotypes. The EAG responses to artificial food in females (**E**) and males (**F**) were measured separately. (Top) Representative EAG recordings. (**G, H**) EAG responses to body extracts in fed and starved flies of different genotypes. The EAG responses to body extracts of females (**G**) and males (**H**) were measured separately. (Top) Representative EAG recordings. Data are means ± SEM, n=10–12 antennae for each condition. Different letters above the error bars indicate significant differences (one-way ANOVA followed by Tukey's multiple comparisons test; p<0.05).

The online version of this article includes the following source data and figure supplement(s) for figure 3:

**Source data 1.** Sk-SkR1 signaling is required for changes in behavior and olfactory sensitivity induced by starvation.

**Figure supplement 1.** Behavioral phenotypes of Sk heterozygous mutants.

**Figure supplement 1—source data 1.** Behavioral phenotypes of Sk heterozygous mutants.

**Figure supplement 2.** Electroantennogram (EAG) responses of starved flies with different genotypes to food relevant odors and sex pheromones at three concentrations.

**Figure supplement 2—source data 1.** EAG responses of starved flies with different genotypes to food relevant odors and sex pheromones at three concentrations.

WT flies but unchanged in starved *sk⁻/⁻* flies (*Figure 4B*). OR67d.3 and OR74a mRNA levels were similar in the satiated and starved WT flies. The differentially expressed genes identified by RNA-Seq were further validated by real-time reverse-transcription PCR (qRT-PCR) (*Figure 4B*).

## Sulfakinin regulates the expressions of starvation-induced ORs that detect food volatiles and starvation-suppressed ORs that detect pheromones

The OR genes identified in the RNA-Seq experiments were expressed in Flp-In T-REx 293 cells followed by the evaluation of functional responses to odorants using calcium mobilization assays. Tests with single high doses of volatiles representing food or sex pheromones allowed the functional categorization of ORs based on odorant specificities (*Figure 4C*). OR10a was activated by ethyl benzoate (EBE), OR7a.8 was activated by ethyl butyrate (EBU) or diethyl maleate (DM), and OR7a.4 was activated

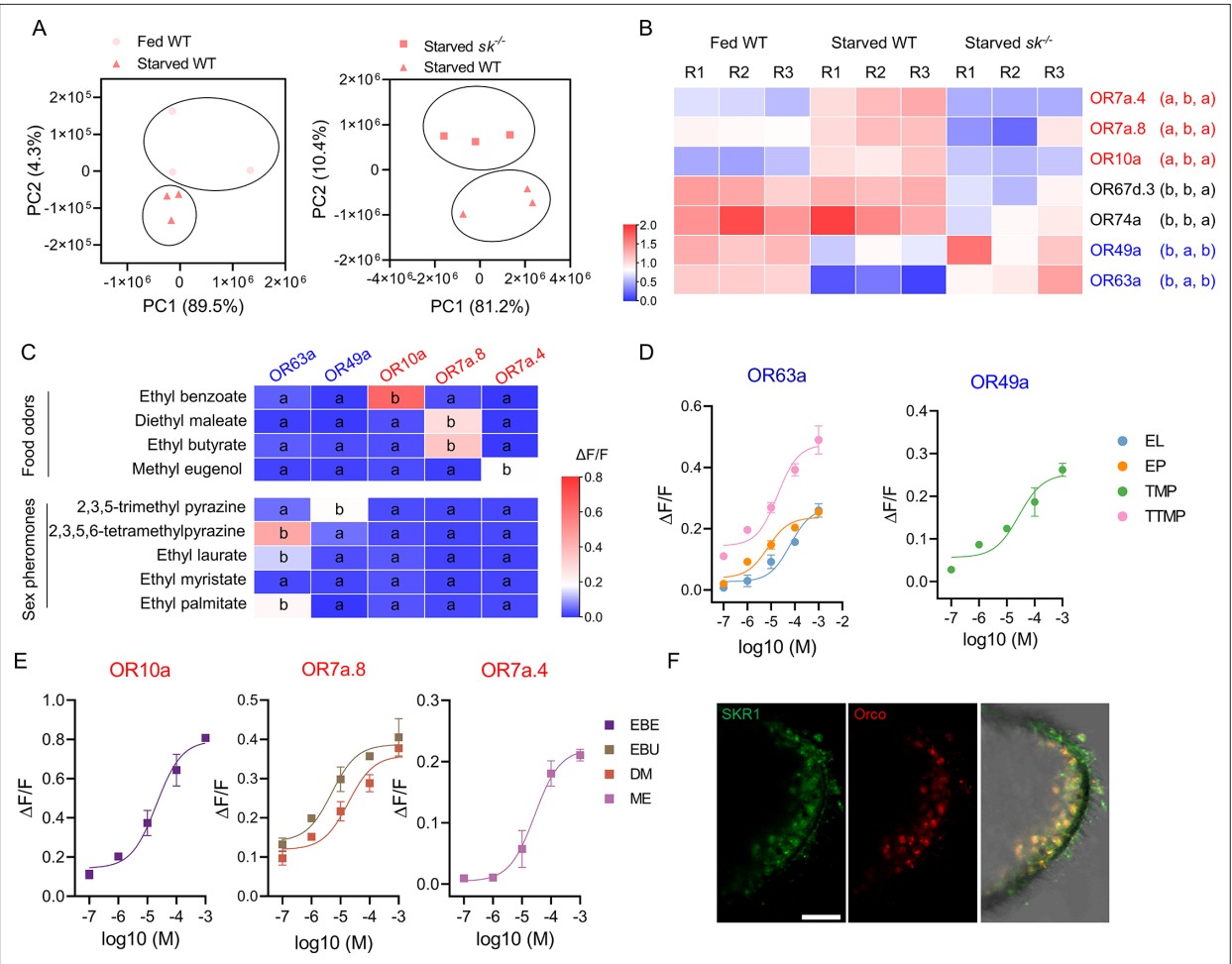

**Figure 4.** The transition of olfactory responses is associated with the expression of different sets of odorant receptors (ORs) in OR neurons induced by Sk-SkR1 signaling. (**A**) Principal component analysis (PCA) using differentially expressed genes obtained from pairwise comparisons between different treatments. (**B**) The expression profiles of the candidate OR genes in wild-type (WT) and $sk^{-/-}$ flies determined by qRT-PCR. The OR names with red fonts represent the set of ORs for food odor and blue fronts represent the set of ORs for sex pheromone. Data are mean relative expression levels ± SEM. R1–R3 represent biological replicates. Different lowercase letters indicate significant differences between treatments (one-way ANOVA followed by Tukey's multiple comparisons test; p<0.05). (**C**) Quantification of calcium levels following the response of candidate ORs to sex pheromones and food odors at a concentration of $10^{-4}$ M (n=3−6). Different lowercase letters indicate significant differences between treatments (one-way ANOVA followed by Tukey's multiple comparisons test; p<0.05). (**D**) Dose–response curves of candidate ORs to sex pheromone components such as ethyl laurate (EL), ethyl palmitate (EP), 2,3,5-trimethyl pyrazine (TMP), and 2,3,5,6-tetramethylpyrazine (TTMP). (**E**) Dose–response curves of candidate ORs to food odorants such as ethyl benzoate (EBE), ethyl butyrate (EBU), diethyl maleate (DM), and methyl eugenol (ME). In both cases, n=5. (**F**) Co-localization of Orco (red) and SkR1 (green) neurons in *B. dorsalis* antennae. Scale bars = 50 µm.

The online version of this article includes the following source data for figure 4:

**Source data 1.** The transition of olfactory responses is associated with the expression of different sets of ORs in OR neurons induced by Sk-SkR1 signaling.

by methyl eugenol (ME). The responses to these food odor volatiles were all dose-dependent, with OR10a showing the strongest response to EBE (***Figure 4E***). In contrast, OR63a and OR49a showed dose-dependent responses to sex pheromone components but not to food odor volatiles (***Figure 4C and D***). Therefore, the upregulated *OR* genes in starved WT flies, *OR7a.4*, *OR7a.8,* and *OR10a*, were activated by the food volatiles, while downregulated genes, *OR49a* and *OR63a*, were activated by pheromonal components. Finally, we determined whether *SkR1* was expressed in OR neurons by testing the antennae for immunoreactivity to SkR1 and odorant receptor co-receptor (Orco) antisera. A subset of clustered OR neurons showed immunoreactivity to SkR1 (***Figure 4F***), indicating that sulfakinin acts directly on SkR1-positive OR neurons.

## Discussion

Dynamic changes in the olfactory sensitivity of insect antennae are associated with behavioral transitions in insects. Sensory inputs via olfactory perception often trigger certain behaviors, including foraging and mating (*Anton and Rössler, 2021*; *Gadenne et al., 2016*). Fine-tuning of the antennal OR repertoire would be an efficient way to co-opt behavioral outputs (*Anton and Rössler, 2021*; *Xu et al., 2024*) as we have shown here for the oriental fruit fly *B. dorsalis*. Changes in odorant sensitivity during transitions between foraging and mating have been reported in previous studies. For example, the antennae of female *D. suzukii* become significantly more sensitive to the host fruit odor isoamyl acetate after mating (*Crava et al., 2019*). Similarly, starved *Aedes aegypti* mosquitoes downregulate the sensitivity of lactic acid receptor neurons in grooved peg sensilla, which are used during host search (*Davis, 1984*; *Christ et al., 2017*), whereas satiated mosquitoes increase the sensitivity of OR neurons in trichoid sensilla to compounds representing the oviposition site (*Siju et al., 2010*). In this study, we determined the mechanisms underlying the changes in antennal sensitivity when switching between foraging and mating behavior. Sk-SkR1 signaling was required to increase the success of foraging and to suppress mating during a period of starvation. The activation of SkR1 on the antennal OR neurons upregulates the expression of ORs that detect food volatiles while suppressing those that detect pheromones.

The coordinated actions of sulfakinin on the antennae and other parts of the peripheral and central nervous system in *B. dorsalis* are not yet fully understood. Sulfakinin is generally known as a satiety signal that reduces food intake and suppresses the digestive system in *D. melanogaster* and a number of different invertebrate species (*Nässel and Wu, 2022*), whereas we found that the Sk-SkR1 signal is required to promote foraging by making the antennae more sensitive to food odorants during starvation. In *D. melanogaster*, sulfakinin inhibits the sugar receptor GR64 in the proboscis and proleg tarsi, thus reducing food ingestion (*Guo et al., 2021*). Our data also support the role of sulfakinin as a satiety signal because feeding rate was increased in $sk^{-/-}$ and $skr1^{-/-}$ null mutants (*Figure 2D*). Sulfakinin was shown to inhibit both male and female sexual behavior in *D. melanogaster* (*Wu et al., 2019*), in agreement with our results. In addition to the previous study showing the roles of central nervous system in Sk-mediated inhibition of sexual behavior in *D. melanogaster*, we described an Sk direct action on the peripheral olfactory system in *B. dorsalis.* The underlying mechanism is the secretion of the mature peptide Dsk-2 (but not Dsk-1) by Dsk-MP1 and Dsk-MP3 neurons, which then triggers downstream neurons expressing SkR1 in the central nervous system to suppress male mating behavior (*Wu et al., 2019*). However, there was a contradictory report showing the role of sulfakinin promoting the receptivity in mating in virgin females (*Wang et al., 2022*). The activity of sulfakinin peptides Sk1 and Sk2, through the receptors SkR1 and SkR2, is therefore likely to be complex, with spatially and temporally dynamic effects in multiple downstream systems.

Although successful foraging and EAG responses to food odors were strongly reduced in both null mutants, there was still a greater response than observed in fed WT controls (*Figure 3A, B, E and F*). This incomplete suppression suggests that other mechanisms, in addition to the main Sk-SkR1 signal, are relevant to the response. Indeed, sNPF and tachykinin signaling have both been described in the modulation of olfactory sensitivity in *D. melanogaster*, with sNPF enhancing sensitivity to food odors and tachykinin inhibiting sensitivity to aversive odors (*Ignell et al., 2009*; *Ko et al., 2015*; *Root et al., 2011*). Furthermore, neuropeptide F (NPF) and SIFamide also regulate foraging behavior at the level of the olfactory circuit (*Beshel and Zhong, 2013*; *Martelli et al., 2017*; *Wang et al., 2013*). Notably, NPF and tachykinin also regulate mating behavior at the peripheral level. Tachykinin inhibits courtship behavior by mediating the perception of an anti-aphrodisiac pheromone in male *D. melanogaster* (*Shankar et al., 2015*). NPF neurons are also required to detect female sex pheromones (*Gendron et al., 2014*; *Kuo et al., 2012*). Similarly, SIFamide also regulates mating behavior, although it is not yet clear whether it acts via the olfactory circuit (*Sellami and Veenstra, 2015*; *Terhzaz et al., 2007*). We have not yet investigated the roles of other neuropeptides in *B. dorsalis* foraging and mating behavior, but we found that starvation-induced the neuropeptide receptor genes SkR1, sNPFR, and *SIFamideR1* (p<0.05). In addition, tachykinin, SIFamide, and ion transport peptide (ITP)-like peptide were also induced by starvation, although the increase was not statistically significant. Therefore, other neuropeptides in addition to sulfakinin are likely to be involved in the switch between different sets of ORs in the antenna as a means to control behavior.

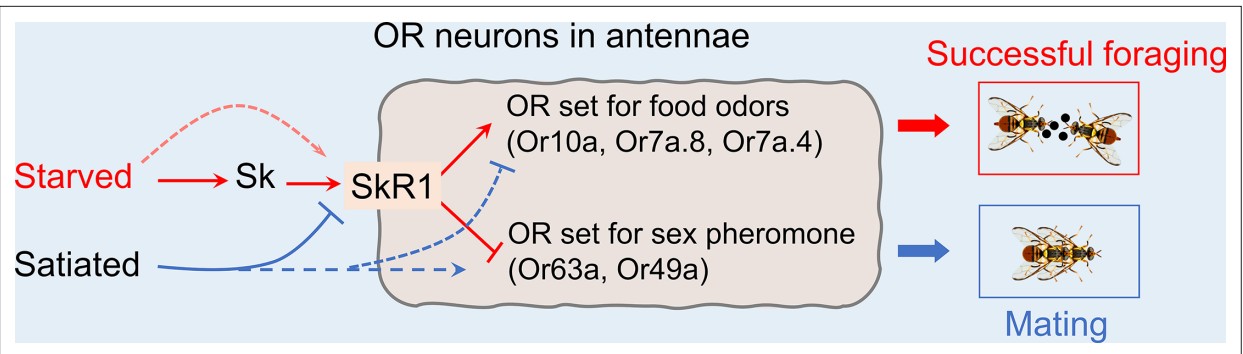

**Figure 5.** Schematic representation of peripheral olfactory remodeling in odorant receptor (OR) neurons of the antenna that arbitrate between mating and foraging behavior in *B. dorsalis*. Starvation increases the abundance of SkR1 in the OR neurons (red arrows) and SkR1 signaling induces the expression of genes encoding ORs that sense food odors, resulting in successful foraging. Satiation suppresses *SkR1* expression and induces the expression of genes encoding ORs that sense opposite-sex pheromones, leading to successful mating (blue arrows). Dashed arrows represent the additional possible pathways that have not been tested in this study, but not excluded in the model. Please see the 'Discussion' for details of additional possible factors modulating odorant sensitivity relevant to satiety.

In summary, we found that the neuropeptide sulfakinin mediates the transition between foraging and mating behavior by activating the receptor SkR1 and reprogramming the peripheral OR repertoire. We propose a working model to characterize this mechanism, in which Sk-SkR1 regulates the expression of different sets of ORs, leading to the behavioral switch (*Figure 5*). The modulation of peripheral sensory systems by sulfakinin, coordinating with the central neural circuit also controlled by sulfakinin, is the major trigger for a successful behavioral switch. We have thus provided new insight into the control of behavior by sulfakinin in *B. dorsalis*, which is one of the most widely distributed, fecund, and invasive insect pests to threaten global agriculture (*Wan and Yang, 2016*; *Zhao et al., 2024*).

## Materials and methods

**Key resources table**

| Reagent type (species) or resource | Designation | Source or reference | Identifiers | Additional information |
|---|---|---|---|---|
| Genetic reagent (*Bactrocera dorsalis*) | sk⁻/⁻ | Wang Lab, Southwest University, this paper | | See Material and methods/CRISPR/Cas9-mediated mutants |
| Genetic reagent (*B. dorsalis*) | skr1⁻/⁻ | Wang Lab, Southwest University, this paper | | See Material and methods/CRISPR/Cas9-mediated mutants |
| Cell line (*Homo sapiens*) | Flip-In T-REx293 cell line | Park Lab, Kansas State University | Cat# R78007 | |
| Antibody | Rabbit polyclonal anti-DSK | Zhou Lab, Chinese Academy of Science | | IHC (1:1000) |
| Antibody | Rabbit polyclonal anti-BdSkR1 | Wang Lab, Southwest University, this paper | | IHC (1:500) |
| Antibody | Rabbit polyclonal anti-BdOrco | Wang Lab, Southwest University, this paper | | IHC (1:200) |
| Antibody | Goat anti-rabbit polyclonal, Alexa Fluor 488 | Cell Signaling Technology | Cat# 4412S | IHC (1:1000) |
| Antibody | Goat anti-mouse polyclonal, Alexa Fluor 647 | Cell Signaling Technology | Cat# 4410S | IHC (1:1000) |

*Continued on next page*

*Continued*

| Reagent type (species) or resource | Designation | Source or reference | Identifiers | Additional information |
|---|---|---|---|---|
| Software, algorithm | ImageJ | National Institutes of Health | https://imagej.nih.gov/ij/ | ImageJ |
| Software, algorithm | Prism 8 | GraphPad | https://www.graphpad.com/ | ImageJ |

## Insects

The WT oriental fruit flies were originally collected from Hainan province, China, in 2008 and were reared in the laboratory (27.5 ± 0.5°C, 75 ± 5% relative humidity, 14 h photoperiod). Larvae and adults were fed with artificial food as previously described (*Shen et al., 2013*). Artificial food for adults, consisting of yeast powder, honey, sugar, ascorbic acid, and water, was also used as a food source for foraging behavior assays.

## Behavioral assays

Virgin females and males, 3-day-old, were placed each in groups of 25 flies and reared at 27.5°C and 60% humidity, and 9-day-old flies were subjected to behavioral assays. Food consumption rates of individual flies were measured by the amount Bromophenol Blue dye (BPB) in the fly after the food including 0.1% (BPB) was provided for 1 h for ad libitum feeding, which is long enough time to allow satiation to occur. BPB was measured from decapitated and homogenized bodies in 1 ml phosphate-buffered saline (PBS) buffer with 1% Triton X-100. Absorbance of the supernatant after centrifugation (13,000 rpm) for 3 min was measured at 614 nm on a xMarkerTM Microplate Spectrophotometer (Bio-Rad). Flies were starved for 12 h prior to the experiments with only water. Foraging behavior assay was performed according to previously published methods with modifications (*Li et al., 2022*). In brief, single flies were introduced into chambers (100 mm diameter × 10 mm height) and acclimatized for 20 min. The foraging behavior was recorded using a camera (Sony HDR CX405) for 15 min at 30 fps. Successful foraging was counted when the fly stayed 5 s or longer within the 5 mm radius of the center where 10 μl of artificial food was placed. The experiments consisted of the assays of three biological replications for each experimental condition. The behavior was analyzed by EasyFlyTracker (*Qu et al., 2021*) with the default parameters.

Mating behavior assay was conducted in the courtship chambers (diameter: 3.5 cm; height: 1.5 cm per layer). The virgin females and males were cold anesthetized at −20°C for 1 min and introduced into the courtship chamber separated by a removable transparent film. After 1 h acclimation, the film was removed to allow the paired flies to get in contact, and mating behavior was recorded by a video camera for 15 min. We measured the level of courtship in mutant males and receptivity in mutant females respectively. For courtship assay, a mutant male ($sk^{-/-}$ or $skr^{-/-}$) and a WT virgin female were introduced into the chambers. Courtship rate is the percentage of the mutant males performs attempted copulation, which was used to measure mutant courtship level. For female receptivity assay, a mutant female ($sk^{-/-}$ or $skr^{-/-}$) and a WT virgin male were introduced into the chambers. Copulation rate is the percentage of the mutant females had engaged in copulation, which is used to measure the mutant female receptivity. The number of flies that had performed attempted copulation or engaged in copulation by the end of each 1 min interval were summed within 15 min and plotted as a percentage of total flies for each time point.

## Electroantennograms

EAGs were used to record the stimulus-dependent changes in the electrical potential summed over the whole antenna. To test volatiles were food odor or body extract of each sex on the flies conditioned for satiated and starved. Different genotypes of the flies were wild-type, Sk mutant, and SkR1 mutants. Female body extracts were tested by male antennae and male body extracts by female antennae (*Supplementary file 1*). Body extracts were prepared by washing 15 9-day-old virgin flies of each sex in 1 ml of hexane for 4 h (*Chen et al., 2023*; *Dweck et al., 2015*; *Ebrahim et al., 2023*) at room temperature (RT) with gentle shaking.

For EAGs, similar to previous study (*Li et al., 2022*), the head (including the antennae) of fly was excised and a glass capillary filled with 0.9% NaCl was inserted into the cut end as the reference

electrode. The tip of an antenna was placed into another glass capillary filled with 0.9% NaCl as the recording electrode. The reference electrode was connected to a grounding electrode, while the recording electrode was connected to a 10× high-impedance DC amplifier (Syntech, Hilversum, the Netherlands). The signal was then sent to an analog/digital converter (IDAC-4, USB, Syntech) and transferred to a computer. Finally, the data were analyzed and saved by using GC-EAD software (Syntech). A filter paper strip (3 × 0.75 cm), loaded with 10 µl of the odor solution or body extract, was placed into a pipette and delivered a stimulus into a constant, humidified airstream flowing at 100 ml/min by a stimulus controller (Syntech). Two stimuli were made for 0.5 s duration with a 30 s interval. Average EAG response, as shown by mV changes, was used for the calculation. EAG for sex pheromones and body surface extracts were measured from 21:00 to 23:00, and food odor was measured from 9:00 to 11:00.

## Quantitative real-time reverse transcription PCR

Antennal tissues from flies of different genotypes with different treatments were dissected and immediately stored at −80°C. Total RNA was extracted with Trizol Reagent (Invitrogen) following the manufacturer's protocol. First-strand cDNA was synthesized using the PrimeScript 1st Strand cDNA Synthesis Kit (Takara, Dalian, China) and specific primers designed using the online software, Primer 3 (http://primer3.ut.ee/, *Supplementary file 2*). Quantitative real-time reverse transcription PCR (qRT-PCR) was performed using a CFX Connect Real-Time System (Bio-Rad, Hercules, CA) with NovoStart SYBR PCR SuperMix kit (Novoprotein, Shanghai, China). Two reference genes (*Shen et al., 2010*), α-tubulin (GenBank: GU269902) and RPS3 (GenBank: XM_011212815), were used selected to normalize the expression levels for the calculation in qBASE (*Hellemans et al., 2007*). Each experiment comprised three independent biological replicates and two technical replicates.

## CRISPR/Cas9-mediated mutants

In vitro synthesis of single-guide RNA (sgRNA) was performed as previously described (*Li et al., 2022*). In brief, the suitable target sites of *Sk* and *SkR1* genes were searched by CasOT (*Xiao et al., 2014*), and the corresponding sgRNAs were synthesized in vitro and purified using the GeneArt Precision gRNA Synthesis Kit (Invitrogen, Thermo Fisher Scientific, Waltham, MA). We designed and synthesized one sgRNA for the *Sk* gene (*Figure 2—figure supplement 2*) and two sgRNAs for the *SkR1* gene (*Figure 2—figure supplement 3*).

The sgRNA and cas9 protein (Invitrogen) were gently pipetted and mixed in nuclease-free water to form a mixture of 400 ng/µl sgRNA and 300 ng/µl Cas9 protein. Eggs collected within 20 min of oviposition were injected within 2 h with the mixture of sgRNA and Cas9 protein using an IM300 Electric Microinjector (Narishige, Tokyo, Japan). The injected eggs were incubated at 27.5 ± 0.5°C and 75 ± 5% relative humidity for hatching.

## Mutant screening and off-target analysis

Heterozygous G1 offspring were obtained using 9-day-old virgin adults (G0) crossed to wild-type opposite sex. Suitable mutant G1 ($sk^{+/-}$ and $skr1^{+/-}$) were backcrossed to WT flies for at least eight generations, then homozygous mutant flies ($sk^{-/-}$ and $skr1^{-/-}$) were obtained by self-crossing. Genomic DNA was extracted from individual flies (G0) or a single hind leg (G1 and later generations) by incubating tissue samples in 30 µl InstaGene Matrix (Bio-Rad) at 56°C for 30 min and inactivating the enzyme at 100°C for 10 min. Fragments of 300–500 bp spanning the sgRNA target sites were amplified using the primers (*Supplementary file 2*). The fragments were genotyped using the QIAxcel Advanced System (QIAGEN, Hilden, Germany), and potential edited products were directly sequenced or cloned into vector pGEM-T Easy (Promega, Madison, WI) for Sanger sequencing to identify the mutation.

To assess off-target effects, CasOT was used to search potential off-target sites for the *Sk* and *SkR1* genes. All loci with one potential mismatch allowed in the seed region and up to three potential mismatches in the non-seed region were searched on the genome sequence (https://db.cngb.org/search/project/CNP0003192/). Two potential off-target sites were found for *Sk* gene and three for *SkR1* gene. PCR amplification and sequencing of fragments including potential off-target sites were performed to check for mutations. In examination of the potential off-target sites with >4 bp mismatches to the sgRNA sequences, no mutations were found in the off-sites (*Supplementary file 3*).

## Immunohistochemistry

Rabbit anti-BdSkR1 antibody was generated by immunization with the peptide N'-RQGLRKCRD-QWP-C'. Mouse anti-BdOrco antibody was raised by immunization with two peptides, N'-GTNPNGL-TRKQE-C' and N'-HWYDGSEEAKTF-C'. Peptides were synthesized and conjugated with keyhole limpet hemocyanin through additional cysteine residues at their N termini by Zoonbio Biotechnology (Nanjing, China). Rabbit anti-BdSk antibody was a gift from Dr. Chuan Zhou (*Wu et al., 2020*). The antisera used to recognize Sk peptide were raised in New Zealand white rabbits using the synthetic peptide N'-GGDDQFDDYGHMRFG-C.

Immunohistochemistry was performed as previously described (*Li et al., 2022*). In brief, whole brains of 9-day-old flies were dissected in PBS, fixed overnight in 4% paraformaldehyde (PFA) in PBS at 4°C, and washed (3 × 15 min) in PBST (0.5% Triton X-100 in PBS). The tissues were blocked in 5% normal goat serum for 20 min at RT, then incubated in primary antibody for 48 h at 4°C. After three washes in PBST (3 × 15 min), the tissues were then incubated in secondary antibody overnight at 4°C. After washing in PBST again, the tissues were mounted on a slide with 100% glycerol for imaging. Adult antennae were dissected and embedded in Tissue-Tek O.C.T. Compound (Sakura, Tokyo, Japan). Sections (18 µm) were prepared using a Thermo HM525 NX microtome at −22°C and then fixed in 4% PFA in PBS at RT for 2 h. The subsequent procedures were as described above for the brain tissues. Primary antibody used: rabbit anti-Sk (1:1000), rabbit anti-SkR1 (1:500) and mouse anti-Orco (1:200). Secondary antibodies used: goat anti-rabbit IgG conjugated to Alexa Fluor 488 (1:1000, Cell Signaling Technology) and goat anti-mouse IgG conjugated to Alex Fluor 647 (1:1000, Cell Signaling Technology). Samples were imaged using an LSM780 confocal microscope (Carl Zeiss, Jena, Germany) and processed with ImageJ software (*Schneider et al., 2012*).

## RNA-seq analysis

Total RNA was extracted from the head epidermal tissues, including the antennae, of fed WT flies, starved WT flies, and starved $sk^{-/-}$ flies, using TRIzol reagent (Invitrogen). Three replicates were performed. Library construction and sequencing was performed by Novogene with Illumina HiSeq2000 platform (Novogene Bioinformatics Technology Co. Ltd, Beijing, China). The raw data (PRJNA1243736) were analyzed after filtering the low-quality sequences by fastp with the default parameters (*Chen et al., 2018*). Sequences were aligned to the *B. dorsalis* genome (https://db.cngb.org/search/project/CNP0003192/) using Hisat2 (version 2.2.1; *Kim et al., 2015*; *Yang et al., 2023*). Samtools (version 1.7) was used to convert sam files to bam files in the process (*Li et al., 2009*). Finally, to calculate the count, we use featureCounts (version 2.0.1; *Liao et al., 2014*). The expression level of genes from the RNA sequencing was normalized by the FPKM method. Differential expression analysis was performed using the DESeq2 R package (1.16.1; *Anders and Huber, 2010*).

## Functional heterologous expression of ORs and calcium imaging

Heterologous expressions of ORs were made in Flip-In T-REx293 cell line (Thermo Fisher), which were cultured in Dulbecco's modified Eagle medium (11995065, Gibco, Thermo Fisher Scientific) containing 10% fetal bovine serum (10091148, Gibco) and 1% penicillin-streptomycin-Amphotericin B solution (C0224-100ml, Beyotime) with 5% $CO_2$ at 37°C. Cells for transfection were seeded onto a 96-well black well plate (165305, Thermo Fisher Scientific). Approximately 50–70% confluent cells were transfected by 2 µg pcDNA3.1 plasmid containing respective olfactory receptors with BdOrco of *B. dorsalis* using *Trans*IT-LT1 Transfection Reagent (MIR 2300, Mirusbio) for 24 h. The cell line from Thermo Fisher (Cat#R78007) was generously supplied by Dr. Yoonseong Park (Kansas State University), which is authenticated by STR profiling. Cells were tested and found not to be contaminated with mycoplasma.

For calcium imaging experiment, the culture media were replaced with 100 µl of 2.5 µM Fluo-4 AM (F14201, Invitrogen) and incubated at 37°C with 5% $CO_2$ for 1 h. Subsequently, after another 30 min of incubation at RT, the culture media containing Fluo-4 were removed, and the cells were washed three times with Hanks' balanced salt solution (HBSS; 14065056, Gibco, Thermo Fisher Scientific) in order to fully remove the Fluo-4 residue. Then, 99 ul of HBSS was added to each well and plate was placed in the dark at RT to be tested. Finally, 1 µl of test odor was added to each well and fluorescence immediately was measured for at least 80 s. Finally, images were captured using a Zeiss Observer 7 inverted microscope (Carl Zeiss, Jena, Germany). Test odors were dissolved in dimethylsulfoxide (DMSO, ST038-100ml, Beyotime) and prepared five final concentrations: $10^{-7}$,

$10^{-6}$, $10^{-5}$, $10^{-4}$ M, and $10^{-3}$ M. the fluorescent ratio ($\Delta F/F$) was calculated as peak intensity divided by basal intensity (intensity before ligand perfusion). For all experiments, the response to DMSO was used as a negative control.

## Statistical analysis

For qRT-PCR result analysis, relative expression levels were calculated using qBase (*Hellemans et al., 2007*) based on the expression of the reference genes. Statistical analysis was performed using GraphPad Prism and indicated in each figure legend. Data were first verified for normal distribution using the Shapiro–Wilk normality test. One-way ANOVA, followed by Tukey's multiple comparisons (if normal distribution) and Kruskal–Wallis and post hoc Mann–Whitney *U* tests (if non-normal distribution), was used for comparisons among multiple groups. Values are reported as means ± SEM.

## Acknowledgements

This research was supported by National Key R&D Program of China (2022YFC2601000), the National Natural Science Foundation of China (32072491, 31772233), 111 Project (B18044), and the China Agriculture Research System of MOF and MARA. We are grateful to Dr. Chuan Zhou and Dr. Feng-ming Wu (Institute of Zoology, Chinese Academy of Sciences) for the DSK antibody. We thank Dr. Tao Wang (Institute of Zoology, Chinese Academy of Sciences) for her graphical assistance for this manuscript.

## Additional information

### Funding

| Funder | Grant reference number | Author |
|---|---|---|
| National Natural Science Foundation of China | 32072491 | Hong-Bo Jiang |
| National Key Research and Development Program of China | 2022YFC2601000 | Hong-Bo Jiang |
| National Natural Science Foundation of China | 31772233 | Hong-Bo Jiang |
| 111 Project | B18044 | Jin-Jun Wang |
| China Agriculture Research System of MOF and MARA | | Jin-Jun Wang |

The funders had no role in study design, data collection and interpretation, or the decision to submit the work for publication.

### Author contributions

Hong-Fei Li, Conceptualization, Investigation, Visualization, Methodology; Bao Dong, Investigation, Visualization, Methodology; Yuan-Yuan Peng, Investigation, Methodology; Hao-Yue Luo, Xiao-Lan Ou, Zheng-Lin Ren, Investigation; Yoonseong Park, Supervision, Writing – review and editing; Jin-Jun Wang, Conceptualization, Resources, Writing – review and editing; Hong-Bo Jiang, Conceptualization, Supervision, Funding acquisition, Writing – original draft, Writing – review and editing

### Author ORCIDs

Hong-Fei Li ⬤ https://orcid.org/0000-0003-3631-5525
Jin-Jun Wang ⬤ https://orcid.org/0000-0002-8777-5268
Hong-Bo Jiang ⬤ https://orcid.org/0000-0002-7314-7391

Joint Public Review: https://doi.org/10.7554/eLife.100870.4.sa1
Author response https://doi.org/10.7554/eLife.100870.4.sa2

## Additional files

### Supplementary files
Supplementary file 1. Chemicals used for electroantennograms.

Supplementary file 2. Primer sequences used in this study.

Supplementary file 3. Analysis of off-target effects of mutants.

MDAR checklist

### Data availability
RNAseq data have been deposited in NCBI under accession code PRJNA1243736.

The following dataset was generated:

| Author(s) | Year | Dataset title | Dataset URL | Database and Identifier |
|---|---|---|---|---|
| HF Li | 2025 | Bactrocera dorsalis (oriental fruit fly) | https://www.ncbi.nlm.nih.gov/bioproject/PRJNA1243736 | NCBI BioProject, PRJNA1243736 |

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
