## [Editor Report · eLife Assessment]

This **important** work investigates the mechanism that underlies the switch between feeding and mating behaviors in the oriental fruit fly, *Bactrocera dorsalis*. Using a variety of approaches, the authors show that this switch is mediated by the neuropeptide, sulfakinin, acting peripherally through the sulfakinin receptor 1 to regulate the expression of antennal odorant receptors. The evidence is **solid** in support of the hypothesis that sulfakinin signaling mediates changes in the periphery, although additional sites of action may also contribute to these changes.

---

## [Referee Report · Joint Public Review]

Summary:

The behavioral switch between foraging and mating is important for resource allocation in insects. This study characterizes the role of sulfakinin and the sulfakinin receptor 1 in changes in olfactory responses associated with foraging versus mating behavior in the oriental fruit fly (Bactrocera dorsalis), a significant agricultural pest. This pathway regulates food consumption and mating receptivity in other species; here the authors use genetic disruption of sulfakinin and sulfakinin receptor 1 to provide strong evidence that changes in sulfakinin signaling modulate antennal responses to food versus pheromonal cues and alter the expression of ORs that detect relevant stimuli.

Strengths:

The authors utilize multiple complementary approaches including CRISPR/Cas9 mutagenesis, behavioral characterization, electroantennograms, RNA sequencing and heterologous expression to convincingly demonstrate the involvement of the sulfakinin pathway in the switch between foraging and mating behaviors. The use of both sulfakinin peptide and receptor mutants is a strength of the study and implicates specific signaling actors.

Weaknesses:

The authors demonstrate that SKR is expressed in olfactory neurons, however there are additional potential sites of action that may contribute to these results.

---

## [Author Response]

The following is the authors’ response to the previous reviews

**Joint Public Review:**
Summary:The behavioral switch between foraging and mating is important for resource allocation in insects. This study characterizes the role of sulfakinin and the sulfakinin receptor 1 in changes in olfactory responses associated with foraging versus mating behavior in the oriental fruit fly (Bactrocera dorsalis), a significant agricultural pest. This pathway regulates food consumption and mating receptivity in other species; here the authors use genetic disruption of sulfakinin and sulfakinin receptor 1 to provide strong evidence that changes in sulfakinin signaling modulate antennal responses to food versus pheromonal cues and alter the expression of ORs that detect relevant stimuli.Strengths:The authors utilize multiple complementary approaches including CRISPR/Cas9 mutagenesis, behavioral characterization, electroantennograms, RNA sequencing and heterologous expression to convincingly demonstrate the involvement of the sulfakinin pathway in the switch between foraging and mating behaviors. The use of both sulfakinin peptide and receptor mutants is a strength of the study and implicates specific signaling actors.Weaknesses:The authors demonstrate that SKR is expressed in olfactory neurons, however there are additional potential sites of action that may contribute to these results.
**Recommendations for the authors:**
The authors have addressed most of the issues raised by the reviewers. Below are a few outstanding issues.(1) Lines 68-69 describe "control of B. dorsalis include the use of the behavioral responses to semiochemicals" but does not describe what these responses are or how behavior is modulated.

The sentence was revised as “Control of B. dorsalis include the use of the reproductive and feeding behavioral responses to semiochemicals” (lines 69 in the revision).

(2) Statistical analysis for 9 hour starved females at 5 minutes is missing in Figure 1D and S1.

We had added statistical analysis for 9 hour starved females at 5 minutes in the revised Figures 1D and S1, respectively (lines 578).

(3) The legend in Figure S2 should be revised as it is not clear from the figure which of the odors are food associated odors.

As suggested, we added food odor label in the revised Figure S2 (lines 666).

(4) Line 167: "Therefore, the upregulated OR genes in starved WT flies, OR7a.4, OR7a.8 and OR10a, were activated by the pheromonal components, while down regulated genes, OR49a and OR63a, were activated by food volatiles." Based on the data, this sentence is incorrect - Therefore, the upregulated OR genes in starved WT flies, OR7a.4, OR7a.8 and OR10a, were activated by the food components, whereas downregulated genes, OR49a and OR63a, were activated by pheromonal components."

We are sorry for our mistake. We had corrected it (lines 168-169).

(5) Line 192: "The coordinated action of sulfakinin on mutiple downstreams,..." should be revised to "downstream pathways or tissues" or simply removing "multiple downstream".

As suggested, we removed “multiple downstream”. See line 192.

(6) Reference formatting is inconsistent: see line 207 vs line 208.

We had corrected it as “(Wu et al., 2019)” (lines 207).

(7) Lines 241-244 The broad discussion regarding the evolution and ancestral function of CCK here and the phylogeny in Figure S6 are peripheral to the authors claims.

As suggested, we removed the section and the Figure S6 in the revision.